# Analysis of YOLOv5 and DeepLabv3+ Algorithms for Detecting Illegal Cultivation on Public Land: A Case Study of a Riverside in Korea

**DOI:** 10.3390/ijerph20031770

**Published:** 2023-01-18

**Authors:** Kyedong Lee, Biao Wang, Soungki Lee

**Affiliations:** 1Geo-Information System Research Institute, Panasia, Suwon 16571, Republic of Korea; 2School of Civil Engineering, Chungbuk National University, Cheongju 28644, Republic of Korea; 3School of Resources and Environmental Engineering, Anhui University, Hefei 230601, China; 4Terrapix Affiliated Research Institute, Cheongju 28644, Republic of Korea

**Keywords:** illegal cultivation, YOLOv5, DeepLabv3+, public land, time series

## Abstract

Rivers are generally classified as either national or local rivers. Large-scale national rivers are maintained through systematic maintenance and management, whereas many difficulties can be encountered in the management of small-scale local rivers. Damage to embankments due to illegal farming along rivers has resulted in collapses during torrential rainfall. Various fertilizers and pesticides are applied along embankments, resulting in pollution of water and ecological spaces. Controlling such activities along riversides is challenging given the inconvenience of checking sites individually, the difficulty in checking the ease of site access, and the need to check a wide area. Furthermore, considerable time and effort is required for site investigation. Addressing such problems would require rapidly obtaining precise land data to understand the field status. This study aimed to monitor time series data by applying artificial intelligence technology that can read the cultivation status using drone-based images. With these images, the cultivated area along the river was annotated, and data were trained using the YOLOv5 and DeepLabv3+ algorithms. The performance index mAP@0.5 was used, targeting >85%. Both algorithms satisfied the target, confirming that the status of cultivated land along a river can be read using drone-based time series images.

## 1. Introduction

In 2017, Asan City, South Korea suffered extensive flood damage due to the collapse of an embankment. Accordingly, in 2018 and 2019, the local government studied the conditions of the river sites and conducted intensive crackdowns on illegal cultivation at these sites. These efforts led to the restoration of the river embankment that had been damaged by illegal farming over several years. However, illegal farming cases have recently increased again. Given that crackdowns across a wide range of areas are time consuming and expensive, they become a burden on local governments. A more appropriate method would be to implement monitoring strategies using drones for regular surveillance, which would allow rapid targeted crackdowns. Given that cultivated lands along rivers are relatively small in area but have a high level of plant species richness and diversity, establishing time series learning data for plants and undertaking regular monitoring through an artificial intelligence (AI) model is necessary.

Deep-learning-based methods have been demonstrated to be more accurate than previous techniques and use deep neural network analysis to detect weeds among crops based on large-scale learning datasets and pre-trained models [1]. Li et al. [2] estimated crop yield and biomass by calculating the vegetation index of three crops using hyperspectral images and performing AI-based automatic machine learning. Drone-based images have become one of the main sources of geographical information system data that support decision-making in various fields. GeoAI is a dataset used to train object detection- and semantic segmentation-related models for geospatial data analysis [3]. Li and Hsu [4] analyzed various images, such as satellite- and drone-based images, street view, and geoscience data, and investigated the development of the GeoAI field through machine vision. Luis et al. [5] proposed a road monitoring system capable of recognizing potholes through drone-based images to detect road surface deterioration. By using pattern recognition technology, the effect of reducing road safety accidents was confirmed [5].

The use of drones to automatically obtain images has shown a high level of effectiveness in terms of time and cost [6,7,8]. Aerial image data are collected through a standard remote-sensing technique, namely using a drone with a specific sensor [9,10]. Drones have the advantage of being able to obtain high-resolution images at relatively low altitudes. Hashim et al. [11] integrated vegetation indices and convolutional neural networks through a hybrid vegetation detection framework. Vegetation inspection and monitoring using drone images are time-consuming tasks. The vegetation index has been used to estimate vegetation health and change [12] and has used AI learning data to overcome the limitations of vegetation recognition techniques. Liao et al. [13] proposed a monitoring system that detects beach and marine litter using drones in real time. Xu et al. [14] monitored oceans, water quality, fish farms, coral reefs, and waves and algae using AI learning. Ullo and Sinha [15] conducted research on various environmental monitoring systems for air quality, water pollution, and radiation pollution. To detect litter using drones, researchers have improved the YOLOv2 model [16,17], modified a loss function in YOLOv3, and created a drone-based automated floating litter monitoring system [18,19]. Tsai et al. [20] presented a convolutional neural network-based training model to estimate the actual distance between people in consecutive images. 

There has been considerable investment in AI machine learning and deep-learning algorithms to maximize safety, cost, and optimization in modern industry [21]. Recently, an AI technique was developed that can automatically identify magnetite in a mine using a multi-spectral camera on a drone [22]. Detecting objects is a key step in understanding images or videos collected from drones [23]. These state-of-the-art deep-learning detectors have seen substantial innovations in recent years. Object detection methods mainly detect a single category such as a person [24,25,26]. However, there have been numerous studies on specific object detection. Regarding object detection using YOLOv5, Mantau et al. [27] suggested YOLOv5 and a new transfer learning-based model for analysis of thermal imaging data collected using a drone for monitoring systems. Liu et al. [28] applied the YOLO architecture to detect small objects in drone image datasets, and the YOLO series [29,30,31] played an important role in object and motion detection tasks [32]. The YOLO series detection method [33] has been widely used for detecting objects from drone-based images because of its excellent speed and high accuracy [34]. Existing detection methods are as follows [35,36,37,38,39]: After exploring each image through pre-set sliding windows, features are extracted, and then trained classifiers are used for categorization [38,39]. Wei et al. [40] added the convolutional block attention module to distinguish buildings with different heights from drone-based images. Additionally, to solve the problem of poor detection performance for damaged roads in drone-based images, Liu et al. [41] proposed an M-YOLO detection method. 

In South Korea, analysis of farmland using drones is being actively conducted. Choi et al. [42] targeted small farmlands using drone-based images and confirmed the applicability of cover classification with algorithms, such as DeepLabv3+, Fully Convolutional DenseNets (FC-DenseNet), and Full-Resolution Residual Networks (FRRN-B). Kim et al. [43] demonstrated the potential for effectively detecting farmland in a water storage area through supervised classification based on the Gray Level Co-occurrence Matrix. Lee et al. [44] studied a method for searching for occupied facilities being used without permission on national and public lands using high-resolution drone images. Chung et al. [45] determined the optimal spatial resolution and image size for semantic segmentation model learning for overwintering crops and confirmed that the optimal resolution and image size were different for each crop. Deep learning is widely used for object classification for analyzing the status of land use [46]. Ongoing studies are investigating the use of YOLOv5 to detect offshore drifting waste [47] and marine litter [48], which have recently emerged as key issues. These artificial intelligence learning models have been applied to various fields, showing potential applications in studies on the safety evaluations of reservoirs [49] as well as in studies predicting fine dust concentrations [50].

In this study, we constructed a dataset with a size of 1024 × 1024 pixels by regularly filming the main riversides in Asan City using a drone. Drone shooting was performed at different altitudes, angles, and directions to collect a diverse dataset. To monitor the time series data, regular filming was performed from July to October. Using the data acquired in this way, the cultivated land was annotated with a polygon to build AI learning data. YOLOv5 and DeepLabv3+ algorithms were applied to the learning data that had been periodically acquired, and the performance goal was mAP@0.5 with an index of 0.85.

## 2. Materials and Methods

### 2.1. YOLOv5 

YOLO is an abbreviation of You Only Look Once, which means to detect an object by looking at an image once [29]. This algorithm can detect objects at a speed closer to real time with a deep-learning network structure that simultaneously processes object detection and classification. YOLO can also divide input images into an N × N size grid and perform a classifier on each cell. Based on this, the probability of the grid cell containing an object is calculated, and the object is detected, as shown in Figure 1.

YOLO has an end-to-end integrated structure and obtains multiple bounding boxes and class probabilities at the same time by inferring images once with a Convolutional Neural Network (CNN). With these features, YOLO has several advantages. First, its mAP and speed are more than twice higher than those of other real-time systems; second, because it uses CNN rather than the sliding windows method, it is induced to contextual information, so the learning rate for each class is good; and third, it can learn the expression of generalized objects. As a result, it has a faster detection speed compared to that of Deformable Part Models (DPM) and Regions with Convolutional Neural Network (R-CNN) [29]. Other object detection models use a combination of a preprocessing model and an artificial neural network. The network configuration of YOLO is relatively simple because it is processed by only one artificial neural network as shown in Figure 2. 

YOLOv5 is implemented based on the PyTorch framework, unlike other versions that are based on the Darknet framework, and has a similar structure to YOLOv4, except that it uses a Cross Stage Partial Network to reduce the calculation time, and its inference time is more rapid than that of YOLOv4. Therefore, YOLOv5 can be applied to small-scale embedded and unmanned mobile systems [48]. 

### 2.2. DeepLabv3+

The DeepLabv3+ model has an encoder-decoder structure. The addition of the decoder has improved model performance compared to that of the previous model DeepLabv3 [51]. The encoder comprises a backbone network marked as a deep convolutional neural network (DCNN) and Atrous Spatial Pyramid Pooling (ASPP). The backbone network is a general convolutional neural network and is specialized for segmentation by applying atrous convolution to some measurements. DeepLabv3+ uses either ResNet-101 [52] or Xception as the backbone network. 

ASPP enables more accurate segmentation by obtaining multi-scale features through the convolution of various kernels. The segmentation map is generated by upsampling the output feature maps of the decoder and encoder. To minimize the restoration loss that occurs at this time, the feature map is reconstructed with two 3 × 3 convolutions after connecting with the output feature map of the encoder, as shown in Figure 3.

### 2.3. Mean Average Precision

Mean average precision (mAP) is a metric used to measure object detection accuracy and is the mean of the average precision (AP) of all classes in the database [53]. To obtain the AP, we must first understand the relationship between precision and recall, which can be defined as shown in Figure 4.

True positive is defined as a correct detection by predicting actual targets. False positive is defined as a false detection/false positive by predicting an object that does not exist. False negative is defined as a misdetection because it does not predict the real object. True negative is defined as a correct detection by not predicting non-existent objects. However, it is not used in object detection frameworks and is based on precision-recall. Precision can be calculated as follows:(1)$$Precision=\frac{TP}{TP+FP}=\frac{TP}{all\;detection}.\;$$

*Precision* is the performance of a model to only identify relevant objects and is the percentage of correctly detected objects out of the detected objects. If the model detects 10 out of the 20 ground truths to be detected, but correctly detects seven objects, then the precision is 0.7. *Recall* can be calculated with the following formula:(2)$$Recall=\frac{TP}{TP+FN}=\frac{TP}{all\;ground\;truths}.$$

*Recall* is the performance of a model to find all the correct answers and is the percentage of correctly detected ground truths. In the example above, among the 20 ground truths to be detected, if there are seven correctly detected objects, then the recall is 0.35. Using this, a curve representing precision according to the change in recall can be displayed, and the model performance can be evaluated with this curve. Given that recall values are always between 0 and 1, mAP can be shown as the following formula using the all-point interpolation method [53]:(3)$$AP_{\mathrm{all}}=\sum_{n}\left(R_{n+1}-R_{n}\right)P_{\mathrm{interp}}\left(R_{n+1}\right),$$
(4)$$P_{\mathrm{interp}}\left(R_{n+1}\right)=\underset{\tilde{R}:\tilde{R}\geq R_{n+1}}{\max}P\left(\tilde{R}\right),$$
(5)$$mAP=\frac{1}{C}\overset{C}{\underset{i}{\sum }}APi.$$

### 2.4. Research Methods

To conduct this study, drone images were obtained for each altitude, angle, and direction for the cultivated area along the river. Filming data were collected regularly at the same place for the time series analysis. To improve the learning and training quality, the drone-based images collected were cut to a certain standard (1024 × 1024 pixels). A refinement step was performed by visual inspection to delete poor-quality images such as those with poor focus, poor color, and file damage. The drone images were taken at a 2-cm spatial resolution, and the images were processed to construct a monthly dataset for learning and training. The cultivated land was annotated with polygons in the refined images, data processing was performed, and learning datasets were built through an inspection process. The learning data were evaluated using YOLOv5 and DeepLapv3+ models. Figure 5 shows the overall flow from data collection to model learning.

### 2.5. Study Area 

This study targeted the main river areas of Asan City, Chungcheongnam-do, South Korea. There were numerous cultivated areas from which data were collected in the vicinity of the river. Drone flights and filming were relatively unrestricted in the target area. As shown in Figure 6, we filmed the areas by dividing them into three parts, namely the northern, central, and southern areas. Field crops were cultivated in B1, rice was cultivated in B2, and crops mixed with natural vegetation were cultivated in B3. Through this, an area that could be analyzed using crop patterns and time series data was selected.

### 2.6. Construction of Experimental Data

We used a DJI Phantom 4 RTK drone for data collection. We collected learning data from July, when crops are commonly grown, to October, when harvesting begins. A total of 24 data collection flights were performed for the entire block by filming each target site twice a month for four months. The number of data collection flights for each block are shown in Table 1. To collect a diverse range of data, we combined shooting methods with different altitudes, angles, and directions, as shown in Figure 7.

The data collected were visually inspected to ensure that they were of high quality. During the inspection process, we removed images that were out of focus because of gas vibrations due to air flows, images with noise due to a lack of light sources, and dark images. Images that passed the quality inspection were divided to a 1024 × 1024 size corresponding to a real area of 20 × 20 m using Adobe Photoshop. Images that did not contain cultivated land or did not meet the standards were deleted, as shown in Figure 8.

The refined data were annotated with a polygon according to the shape of the cultivated land using an authoring tool (by Show Tech). For the consistency of the annotation work, only the parts with a certain farming pattern were defined as farmland. In addition, if farmland with different patterns was adjacent, it was separated and annotated as shown in Figure 9. The amount of data collected in each block by collection period is shown in Table 2, and it is classified as a training dataset, validation dataset, and test dataset, as shown in Table 3.

### 2.7. AI Model Accuracy Evaluation Method

To evaluate the accuracy of the learning model, the data were divided into training, validation, and testing sets at a ratio of 8:1:1. The mAP index was used to compare the YOLO and DeepLabv3+ models. mAP is a comprehensive evaluation index that considers precision/recall. To calculate mAP, a value of AP@IoU ≥ 0.5 was set as a true positive. The AP for cultivated land in each image was obtained, and the mAP was calculated using Equation (5) [33].

As was the case for the YOLO model, we could not train the polygon-processed data. Therefore, we extracted the top, bottom, left, and right maximum values of the cultivated land polygons. They were then converted into a bounding box to enable training, as shown in Figure 10.

### 2.8. Experimental Environments

The training device used in the study was a dual graphics processing unit (GPU) given the amount of data to process and the speed needed. Details are provided in Table 4.

### 2.9. Parameter Setting

To compare the training results of each model, it is necessary to fix the number of training iterations of YOLOv5 and DeepLabv3+. Therefore, referring to previous research [45], the number of iterations and batch size for YOLOv5 and DeepLabv3+ were determined as shown in Table 5.

### 2.10. Training and Evaluation 

Cultivated land was searched using training data (80%) with 120,000 datasets, and the precision and recall for each block are shown in Table 6, Table 7 and Table 8.

As a result of the search, precision and recall were the highest for B2, which had many training datasets and clearly differentiated cultivated land. In the case of B3, the number of training datasets was relatively small, and the shape of the cultivated land was similar to the surrounding natural vegetation. Therefore, the precision and recall of the primary data were low. However, over time, as the cumulative number of training datasets increased and the harvest season arrived, the distinction between arable land and natural vegetation became clear, resulting in increased precision and recall.

## 3. Results

### 3.1. Training Results

Given that most of the cultivated land had a certain pattern, it could be confirmed that both models accurately detected the pattern. 

However, in the case of YOLOv5, it was necessary to convert the polygon to a bounding box. A bounding box may include other objects such as native plants because cultivated land is not standardized, as shown in Figure 11. Problems arose in some cases such as some areas of the bounding box being lost during the conversion process or classes being changed. Therefore, it was confirmed that DeepLabv3+, which does not require preprocessing, provided more accurate identification in the case of cultivated land annotated with a polygon.

### 3.2. Analyses

In this study, a dataset of 120,000 farmland areas was constructed, 80% of which was training data, 10% was validation data, and the remaining 10% was test data. mAP values were calculated for each data acquisition period. As a result of calculating the mAP for each block using the YOLOv5 and DeepLabv3+ models, it was found that both models had the highest mAP values in B2. This had a substantial amount of training data, specific patterns, and time series characteristics. In the case of B1, the mAP value was high due to the difference between the pattern specific to field crops and the natural vegetation in Table 9. The change in mAP value according to time series data was relatively small. In the case of B2, the mAP value was relatively high due to the distinct pattern according to the characteristics of the rice cultivation area in Table 10. However, it was confirmed that there was little effect on the time series data. 

In the case of B3, in Table 11, the mAP value was low at the beginning of data collection because it was mixed with native plants. However, the mAP value increased through time series data. Therefore, the reading rate of farmland along the river can be improved through the diversity of training data.

## 4. Discussion

To efficiently classify the cropland in a reservoir area, Kim et al. [43] used the Gray Level Co-occurrence Matrix (GLCM), which is a representative technique used for quantifying texture information, along with Normalized Difference Water Index (NDWI) and Normalized Difference Vegetation Index (NDVI), as additional features during the classification process. They analyzed the use of texture information according to window size for generating GLCM and proposed a methodology for detecting croplands in the studied reservoir area.

In this study, learning data was constructed to find illegal farming activities along the river. As a result, illegal cultivation patterns were identified along the riverside. A large amount of training data was used to exceed the target mAP value. Also, in the case of YOLOv5, which is not suitable for annotation data with polygons, it was a satisfactory achievement to obtain results close to DeepLabv3+. In order to find illegal farming, a large amount of learning data and a high success rate are required. However, it was not analyzed by applying various algorithms, and the analysis of various illegal activities on land other than arable land was not made. Therefore, in the future, we plan to develop learning data on the illegal behaviors of various waste accumulation patterns and conduct research to discover appropriate algorithms by applying various learning algorithms.

## 5. Conclusions

For cultivated land, the shape differs depending on the crop growth period. Therefore, if the data used is only from a certain moment, then the quality of learning can deteriorate. When filming target sites with a drone, the shape or size may differ depending on the altitude and angle. Therefore, a variety of time series learning data are required. Given that cultivated land generally comprises only crops, it is only necessary to pay attention to the crop growth condition. However, in the case of rivers, various plants other than crops grow. Therefore, it is necessary to identify the characteristics of crops and then train the relevant data. To identify these characteristics, a substantial amount of learning data was collected by acquiring drone-based images at different altitudes, directions, and angles.

The YOLOv5 algorithm uses a bounding box as a basis, and in the case of DeepLabv3+, an object is annotated with a polygon. Therefore, a direct comparison cannot be made. However, in this study, we converted a polygon to a bounding box to use the YOLOv5 algorithm. As a result of the training data after annotating cultivated land with an irregular shape, the mAP@0.5 values were 0.91 for YOLOv5 and 0.96 for DeepLabv3+. The learning result using the YOLOv5 algorithm was confirmed to be similar to that using DeepLabv3+. Both algorithms obtained values exceeding the target of 0.85. By comparing these two algorithms using the time series learning data for cultivated land along a river, illegal farming activities could potentially be detected along the riversides. Illegal cultivation patterns along the riverside were identified. It was confirmed that there were various acts of accumulating waste (other than tillage) along the riverside without permission. Therefore, in future, we plan to develop learning data for various patterns of waste accumulation and conduct research to identify an appropriate algorithm by applying various additional learning algorithms.

## Figures and Tables

**Figure 1 ijerph-20-01770-f001:**
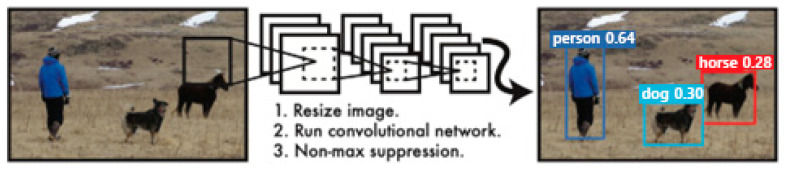
YOLO detection system [29].

**Figure 2 ijerph-20-01770-f002:**
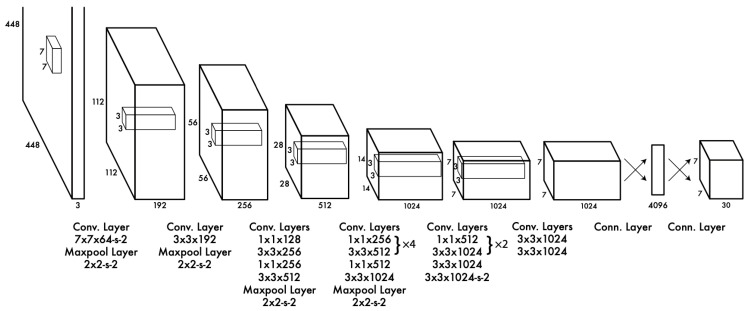
YOLO network architecture [29].

**Figure 3 ijerph-20-01770-f003:**
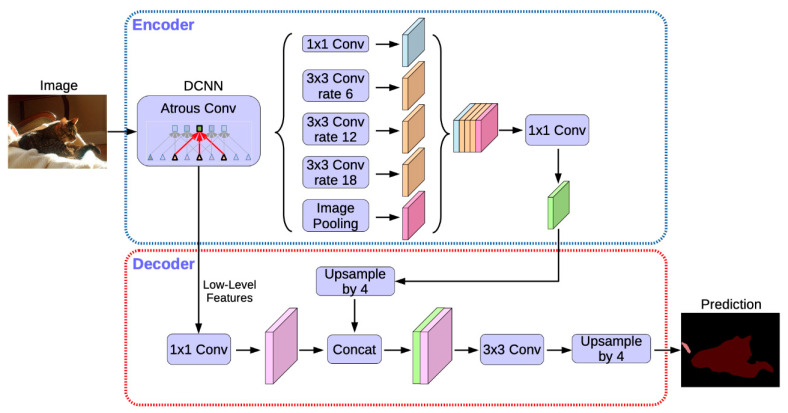
DeepLabv3+ architecture [51].

**Figure 4 ijerph-20-01770-f004:**
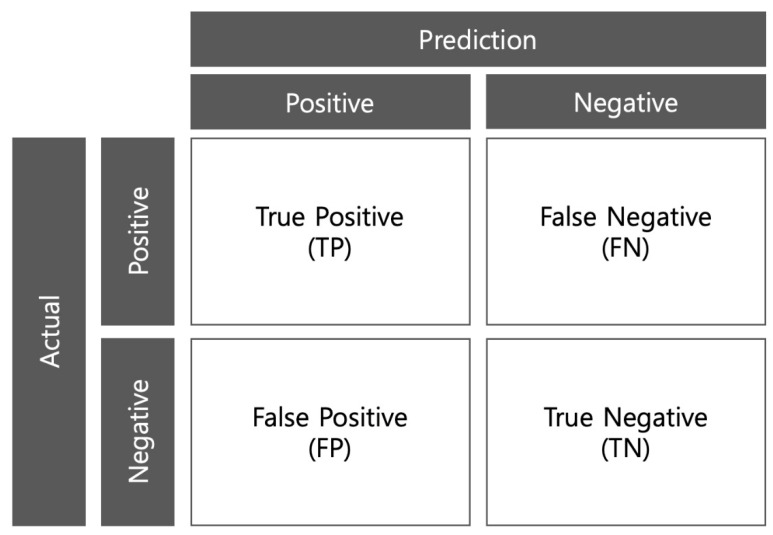
Four factors to obtain the mean average precision index.

**Figure 5 ijerph-20-01770-f005:**
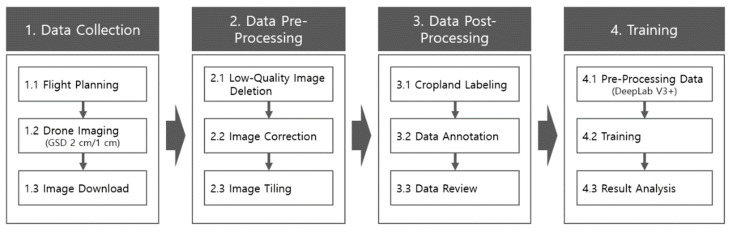
Learning data construction process.

**Figure 6 ijerph-20-01770-f006:**
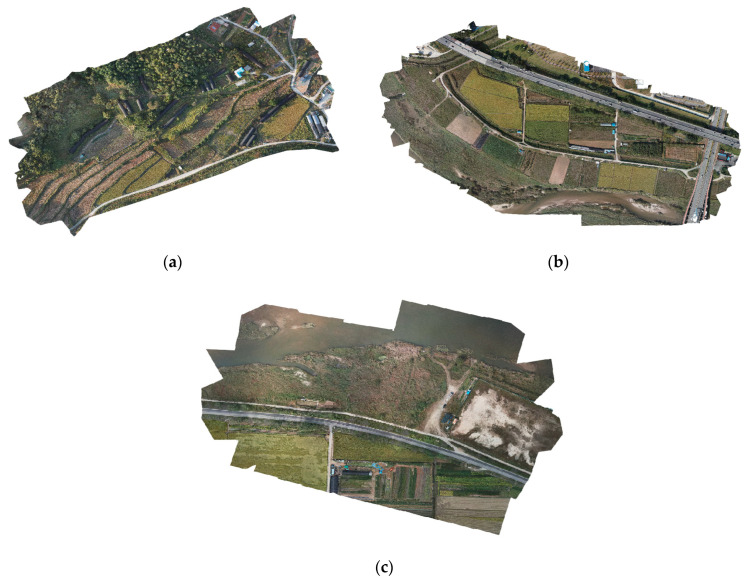
Target sites for data collection in Asan City: (**a**) northern (B1), (**b**) central (B2), and (**c**) southern (B3).

**Figure 7 ijerph-20-01770-f007:**
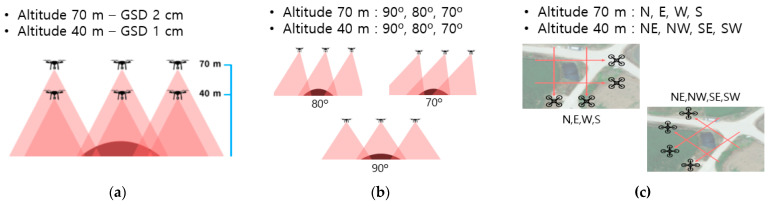
Data collection method: (**a**) photogrammetry per altitude (**b**) photogrammetry per angle (**c**) photogrammetry per direction.

**Figure 8 ijerph-20-01770-f008:**
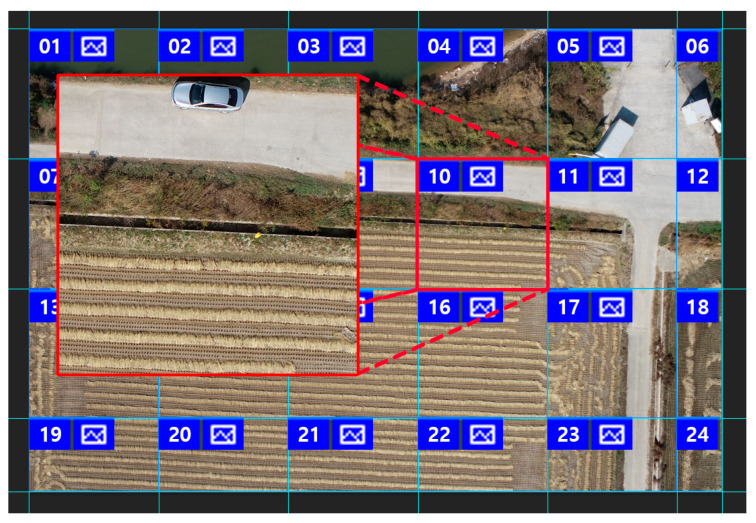
Image division.

**Figure 9 ijerph-20-01770-f009:**
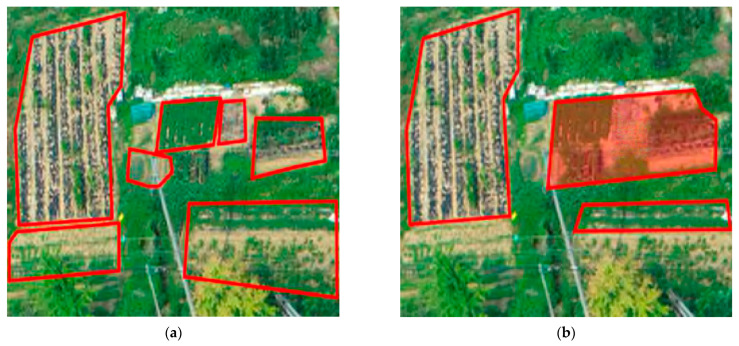
Annotation of cultivated land. (**a**) annotation normal appearance; (**b**) annotation error (red polygon).

**Figure 10 ijerph-20-01770-f010:**
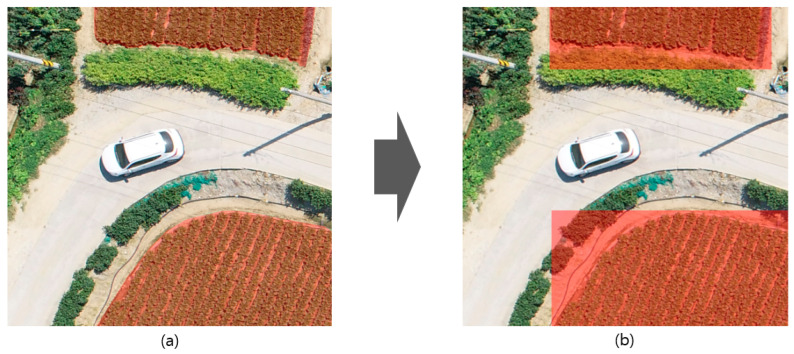
Conversion from polygons to bounding boxes: (**a**) polygon; (**b**) bounding box.

**Figure 11 ijerph-20-01770-f011:**
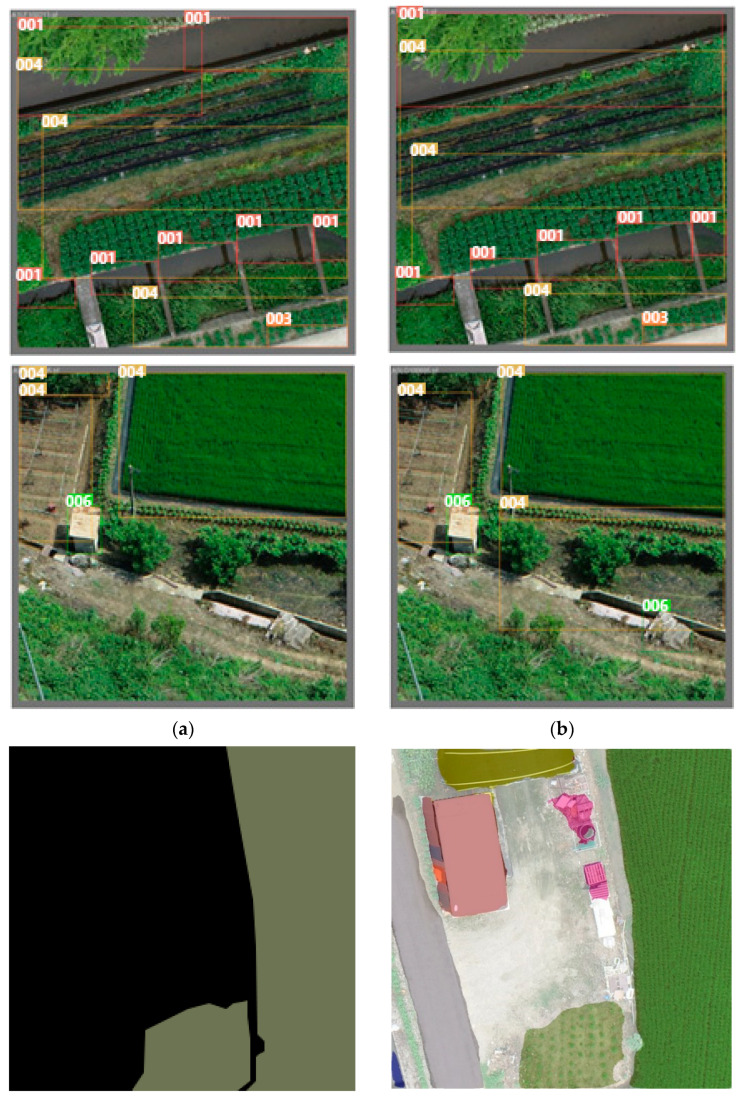
Training results: (**a**) ground truth of YOLOv5; (**b**) prediction of YOLOv5; (**c**) ground truth of DeepLabv3+; (**d**) prediction of DeepLabv3+.

**Table 1 ijerph-20-01770-t001:** Number of data collections.

Target Area	No. of Collections per Month	Area	Time of Collection	Collected Time	Collection Period	Total No. of Collections
(a)	2	94,000 m^2^	10: 00~18: 00	8 h	4 months	8
(b)	2	170,000 m^2^	09: 00~19: 00	15 h		8
(c)	2	37,000 m^2^	11: 00~15: 00	4 h		8

**Table 2 ijerph-20-01770-t002:** Cumulative number of training data collected per block.

Data Collection	No. of Accumulated Data in B1	No. of Accumulated Data in B2	No. of Accumulated Data in B3	Sum
1st	8763	18,023	3214	30,000
2nd	18,072	35,117	6811	60,000
3rd	27,225	53,046	9729	90,000
4th	37,078	67,429	15,493	120,000

**Table 3 ijerph-20-01770-t003:** Number of training dataset.

Block Name	Data Collection	Train Sets (80%)	Validation Sets (10%)	Test Sets (10%)
B1	1st	7010	876	877
	2nd	14,458	1807	1807
	3rd	21,780	2722	2723
	4th	29,662	3708	3708
B2	1st	14,418	1802	1803
	2nd	28,094	3511	3512
	3rd	42,437	5304	5305
	4th	53,943	6743	6743
B3	1st	2571	321	322
	2nd	5449	681	681
	3rd	7783	973	973
	4th	12,394	1549	1550

**Table 4 ijerph-20-01770-t004:** Device environment for training.

Hardware	Performance
CPU	AMD Ryzen Threadripper Pro 5995WX (68 Core, 128 Threads)
GPU	NVIDIA RTX A6000 D6 48GB 2-Way
RAM	ECC 384GB
OS	Ubuntu 20.04.5
Framework	PyTorch

**Table 5 ijerph-20-01770-t005:** Parameter settings for data training.

Parameter	YOLOv5	DeepLabv3+
Epoch	50	50
Batch Size	128	8
Optimizer	SGD	Adamw

**Table 6 ijerph-20-01770-t006:** Cultivated land search results for B1.

Data Collection	Test Data Sets	TP		FP		FN		Recall		Precision	
		YOLO	DLv3+	YOLO	DLv3+	YOLO	DLv3+	YOLO	DLv3+	YOLO	DLv3+
1st	877	684	661	311	347	193	216	78	75	69	66
2nd	1807	1531	1558	337	298	276	249	85	86	82	84
3rd	2723	2336	2548	281	287	387	175	86	94	89	90
4th	3708	3371	3380	259	221	337	328	91	91	93	94

**Table 7 ijerph-20-01770-t007:** Cultivated land search results for B2.

Data Collection	Test Data Sets	TP		FP		FN		Recall		Precision	
		YOLO	DLv3+	YOLO	DLv3+	YOLO	DLv3+	YOLO	DLv3+	YOLO	DLv3+
1st	1803	1689	1680	248	211	114	123	94	93	87	89
2nd	3512	3321	3345	221	178	191	167	95	95	94	95
3rd	5305	5214	5238	192	154	91	67	98	99	96	97
4th	6743	6608	6698	124	89	135	45	98	99	98	99

**Table 8 ijerph-20-01770-t008:** Cultivated land search results for B3.

Data Collection	Test Data Sets	TP		FP		FN		Recall		Precision	
		YOLO	DLv3+	YOLO	DLv3+	YOLO	DLv3+	YOLO	DLv3+	YOLO	DLv3+
1st	322	147	231	84	97	175	91	46	72	64	70
2nd	681	340	488	113	128	341	193	50	72	75	79
3rd	973	762	811	282	198	211	162	78	83	73	80
4th	1550	1337	1470	348	334	213	80	86	95	79	81

**Table 9 ijerph-20-01770-t009:** The mAP results of YOLOv5 and DeepLabv3+ by data collection period for B1.

Data Collection	Training Data Sets	YOLOv5		DeepLabv3+	
		mAP	Training Time (min)	mAP	Training Time (h)
1st	7010	0.88	10	0.90	1
2nd	14,458	0.89	15	0.92	2
3rd	21,780	0.90	20	0.91	3
4th	29,662	0.90	25	0.91	4

**Table 10 ijerph-20-01770-t010:** The mAP results of YOLOv5 and DeepLabv3+ by data collection period for B2.

Data Collection	Training Data Sets	YOLOv5		DeepLabv3+	
		mAP	Training Time (min)	mAP	Training Time (h)
1st	14,418	0.91	15	0.94	2
2nd	28,094	0.92	20	0.96	4
3rd	42,437	0.93	30	0.96	5
4th	53,943	0.93	40	0.95	6

**Table 11 ijerph-20-01770-t011:** The mAP results of YOLOv5 and DeepLabv3+ by data collection period for B3.

Data Collection	Training Data Sets	YOLOv5		DeepLabv3+	
		mAP	Training Time (min)	mAP	Training Time (h)
1st	2571	0.81	5	0.86	0.33
2nd	5449	0.84	8	0.88	0.67
3rd	7783	0.85	10	0.90	1
4th	12,394	0.86	15	0.90	2

## Data Availability

Not applicable.

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
