# Peer review of "Analysis of YOLOv5 and DeepLabv3+ Algorithms for Detecting Illegal Cultivation on Public Land: A Case Study of a Riverside in Korea"

_ijerph, 2023, doi:10.3390/ijerph20031770_

Round 1

Reviewer 1 Report

Lines 105-111 should be placed in the Methods section, while lines 111-115 – in the Conclusions.

Line 184: What visual criteria were used to distinguish poor quality of color and focus?

Line 186: How was monthly dataset constructed? How many images were generally taken?

Just nine lines for Discussion? It is not serious. This section needs complete revision and enlargement, including not only ruminations over the results, but also comparison of the results with similar ones obtained in the studies by other authors, maybe even using another methods for the same purposes.

Reviewer 2 Report

The manuscript compared the performance of two algorithm YOLOv5 and DeepLabv3 in detection of illegal cultivation on public lands. The paper results seem interesting, however, there are some issues that must be addressed before considering it for publication. Therefore, I would suggest major revision.

The specific comments are as follow:

1- L105: what is 1,024*1,024? Please provide the unit.

2- The last paragraph in the introduction must be written again. Normally in last paragraph of the introduction we provide the novelties of our studies and do not mention the final results.

3- In introduction try to expand your literature review specially by mention the papers published in IJERPH. Also to show the feasibility of deep learning models in other fields you can use the following reference as well:

https://doi.org/10.1007/s11356-022-19574-4

doi: 10.2166/hydro.2021.178

4- In materials and methods, the introduction about Yolov5 is not enough. Please provide in depth introduction, how it works, what conv. Layer is? etc.

5- L 183: How to perform refinement step? Which criteria were used to delete those poor quality image?

6- L185: As you used drone in different altitude how to keep the spatial resolution in 2 cm for whole data set?

7- L 226-L 231: Please explain more about this step. How to define the shape, how to classify based on different crop types? What are the techniques? How to distinguish between crop due to tiling or those that grow naturally as your study area is near riverside? All should be clearly mentioned and explained.

Round 2

Reviewer 2 Report

Thank you for your effort.